# Integrated Biological Control of the Sugar Beet Weevil *Asproparthenis punctiventris* with the Fungus *Metarhizium brunneum*: New Application Approaches

**DOI:** 10.3390/pathogens12010099

**Published:** 2023-01-06

**Authors:** Maria Zottele, Martina Mayrhofer, Hannah Embleton, Jürg Enkerli, Herbert Eigner, Eustachio Tarasco, Hermann Strasser

**Affiliations:** 1Department of Microbiology, Leopold-Franzens University Innsbruck, 6020 Innsbruck, Austria; 2AGRANA Research & Innovation Center GmbH, 3430 Tulln an der Donau, Austria; 3Molecular Ecology, Agroscope, 8046 Zurich, Switzerland; 4Department of Soil, Plant and Food Sciences, University of Bari Aldo Moro, 70121 Bari, Italy

**Keywords:** entomopathogenic fungus, pest control, mass application, abundance, mycosis

## Abstract

The mass occurrence of the sugar beet weevil (*Asproparthenis punctiventris,* previously *Bothynoderes punctiventris*) has been endangering sugar beet cultivation in Austria for centuries. Exacerbated by climatic and political changes (warmer, drier spring and limited access to chemical pesticides), new approaches are needed to counter the problem. The aim of our work was to test whether the bioinsecticide *Metarhizium brunneum* Ma 43 (formerly *M. anisopliae* var. *anisopliae* BIPESCO 5/F52) can be used as a sustainable plant protection product against the sugar beet weevil. Our goal was to control the pest in all its development stages through multiple applications. Therefore, GranMet^TM^-P, a granular formulation of *M. brunneum* Ma 43, was applied in spring to establish the fungus in the soil, whereas GranMet^TM^-WP, a liquid formulation of the production strain, was used in early summer on trap ditches and leaves to target the adult weevils. Soil and plant samples as well as weevils were collected during the planting season from the trial sites to evaluate the development of the fungus and the mycosis of the treated weevils. In addition, data on hibernating weevils and their emigration from untreated field sites was collected. In all field sites, the *Metarhizium* spp. abundance increased above the background level (<1000 CFU g^−1^ soil dry weight) after application of the product. With an increasing number of treatments per plot, and thus an increased contact possibility between pest and the fungus, a rise in the mycosis rate was observed. In conclusion, the various *Metarhizium* application strategies, which are already available or in testing, must be implemented to ensure control in both old and new sugar beet fields. *Metarhizium* is a further asset in the successful control of this sugar beet pest.

## 1. Introduction

Since the beginnings of agriculture, there has been a struggle to avoid pest populations in order to guarantee our food supply [1,2]. The more abundant, severe and crop-threatening a pest is, the more effort is needed to solve the pest problem [3]. Over time, however, environmental conditions can change, and some pests may become more frequent or have higher population densities [4], as is the case for the sugar beet weevil (*Asproparthenis punctiventris* Germar, previously *Bothynoderes punctiventris* Germar, Coleoptera: Curculionidae) [5,6]. Problems caused by this pest as well as its control have been the focus of discussions since the 19th century [7]. In Europe, the pest has occurred regularly, especially locally. Ukraine, Croatia, Romania, Bulgaria, Serbia, Hungary, Greece, Austria, Germany, and Italy are among the most affected regions [8,9,10]. It is well known that the weevil prefers warm, dry weather to emerge from the soil and start emigrating [11]. Therefore, the effects of climate change on environmental conditions–particularly in the months of May/June, which are crucial for the weevil’s mass appearance–have led to an increase in incidences of the insect pest and its spread to new regions. Furthermore, changes in the approval and availability of chemical insecticides have further exacerbated the problem [6].

*Metarhizium* spp. Sorokin (Hypocreales: Clavicipitaceae) have long been known as a group of fungi with insecticidal potential. In Europe, the successful commercial use of *Metarhizium* against the June and garden chafer (*Amphimallon solstitiale* L. and *Phyllopertha horticola* L., Coleoptera: Scarabaeidae), the black vine weevil (*Otiorhynches sulcatus* F., Coleoptera: Curculionidae), wireworms (*Agriotes* spp. L., Coleoptera: Elateridae) and the grape phylloxera (*Daktulosphaira vitifoliae* Fitch, Hemiptera: Sternorryncha) is already well-established, and is now being tested to control the Japanese beetle (*Popillia japonica* Newman, Coleoptera: Scarabaeidae; [12]) or the western corn root worm (*Diabrotica virgifera virgifera* LeConte, Coleoptera: Chrisomelidae; [13]).

Biological control of the sugar beet weevil with entomopathogenic fungi has been neglected for the last 100 years. Only a few studies have reported the use of *Beauveria bassiana* against the sugar beet weevil [14,15], *Metarhizium* spp., however, was hardly mentioned. Interestingly, the first fungal infections of the weevil caused by *Metarhizium* spp. were already reported at the end of the 19th century, and the first large-scale treatment with *Metarhizium* was also first carried out against this pest around this time [7,16]. Despite Krassilstchik’s reference [7] to the great potential of this treatment approach, further trials were discontinued in the 20th century. This may be due to the different mode of action of fungal biopesticides compared to the chemically synthesized insecticides. Whereas chemical insecticides are supposed to kill shortly upon contact with the insect, the fungal spores have to adhere, germinate, penetrate the cuticula of the insect and multiply within the insect to have an effect. These key stages in the invasive and developmental processes of the fungus are time-consuming [17]. Nevertheless, there are major advantages in using entomopathogenic fungi against insects. The fungus can be applied to the soil as well as to surfaces. This enables the control not only of the adult weevil but of all developmental stages of the pest, including its larvae. As the spores have a certain resistance to environmental influences, an application does not have to directly coincide with an invasion of the insects [18].

Integrated pest management in sugar beet cultivation is still oriented towards crop-specific requirements. It attempts to continue the use of so-called effective and “intelligent” solutions [19], such as seed dressing with the insecticidal active ingredient clothianidin from the group of neonicotinoids, permitted under an emergency use authorization under Article 53 of Regulation (EC) No 1107/2009. This is what happened in numerous European member state countries in 2022, including Austria, where a temporary emergency authorization for the neonicotinoid treatment of beet seed was granted anew. The further development of integrated pest management and alternative methods or procedures is increasingly in focus in order to reduce the dependence on repeated emergency authorizations. The overall goal should be the reduction of chemical synthetic plant protection products and replacing these with alternative products such as fungal biopesticides. These fulfil the requirement of a sustainable control approach based on natural antagonists. Our strategy is to establish a holistic, alternative control option for all developmental stages of *A. punctiventris* as well as to monitor its emigration to better predict and plan further control measurements.

The major goals of this work were to (i) assess the pathogenicity of the commercially used fungus *Metarhizium brunneum* Petch Ma 43 (previously *M*. *anisopliae* var. *anisopliae* (Metsch.) Sorokin BIPESCO 5/F52) products against the sugar beet weevil; (ii) test different application methods in the sugar beet field for a sustainable establishment of the fungal active agent; (iii) evaluate the rate of fungal infection of the weevils by locally occurring pathogens under optimal fungal growth conditions (25 °C, 60% rH) and (iv) determine the migration of the weevil from the sugar beet fields in Lower Austria.

## 2. Materials and Methods

### 2.1. Trial Sites

Field trials were performed in the most important sugar beet growing area of Lower Austria, which was severely affected by the pest *A. punctiventris* in 2018 and 2019. Seven field sites were chosen (Table 1) to perform duplicates of four different treatment approaches (2.2). In addition, a further trial site was chosen for the spray treatments on sugar beet leaves (field J, n = 24 plots). To determine the natural mycosis rate of the beetles by entomopathogenic fungi, two additional fields were selected that had never been treated with *Metarhizium* spp. The field sites were selected according to routine crop rotation of sugar beet cultivation and predicted weevil infestation (based on monitoring in the previous autumn of pupae/beetles in the fields).

Pot experiments were performed at the University Innsbruck in an open greenhouse. Pots of 40 cm length, 15 cm width and 15 cm depth were filled with a *Metarhizium*-free (the soil was tested for the presence of entomopathogenic fungi before use [20]) standardized soil mixture containing peat, sand and lava granules (ratio 3.5:1:1). All pots were planted with six sugar beet seedlings (organic quality) per replication and left undisturbed until the treatment (n = 4 per treatment). Water was not applied separately. The soil was not further fertilized.

In addition, 13 field sites (further referred to as “Emigration fields”, Appendix A), were selected to evaluate the emigration of weevils (2.3). The fields were chosen based on the results obtained from autumn monitoring of overwintering weevils the previous year, which predicted a high number of migrating weevils. No application was performed on the field sites.

### 2.2. Fungal Application Strategies

The strain *M. brunneum* Ma 43 was applied as granulare (GranMet^TM^-P, Agrifutur Srl., Italy) or spore dispersion (GranMet^TM^-WP, Agrifutur Srl., Italy). The quality of the used products was assessed in the laboratory before application. Assessed parameters were spore density, purity, colonization ability (for granular product) and strain identity [20]. Different application strategies were used to gain insight into advantages and disadvantages of the strategies against the sugar beet weevil.

In a long-term trial, the field sites A and B were treated in autumn 2019, spring 2020 and spring 2021 with 100 kg ha^−1^ GranMet^TM^-P. The application of the product was performed by the farmers directly before the sewing of sugar beet seed using a fertilizer spreader and incorporating it into the soil with a power harrow. With the single annual application, the persistence of the fungus in the treatment areas was assessed.

Field sites C and D were treated with a combination of granular and liquid application. In autumn 2020 and spring 2021, 100 kg ha^−1^ of the granular product GranMet^TM^-P was applied into the soil as previously described. To determine the start of sugar beet immigration, sugar weevil migration was regularly monitored by the authors and farmers. Furthermore, the current weather situation was taken into account, as it influences the behavior of the weevils and the migration time. In addition, the depth at which the weevils were located in the soil was determined by digging. Just before weevil immigration started, the field sites were surrounded by a trap ditch (12 cm width and 40 cm depth, Appendix A) produced with a cable trencher (Cosmeco V1). This trap ditch plus a strip of in total 12 m (due to the working width of the machine) was treated with GranMet^TM^-WP with a spore density of 1 × 10^12^ spores ha^−1^. The liquid formulation was applied with a saddle tank sprayer, which is normally used for the application of liquid chemicals. This dual application should ensure contact of the adult weevils as well as of the newly laid eggs or the emerging larvae with the infectious fungal spores, enabling the control of all developmental stages.

In addition to the large-scale trials, plot trials were performed at three field sites (Field E-G). At the field sites E and F, sugar beet was grown before the application in 2019, whereas field site G was treated before cultivation of sugar beet in 2020. In each field, 12 plots with a size of 5 × 5 m were created, of which half were randomly selected and manually treated with GranMet^TM^-P in spring 2020. The remaining plots were addressed as controls. On each plot, a cage with 2 × 2 m was installed (Appendix A). At field site G, adult *A. punctiventris* (n = 20) were introduced into each cage. In all other fields, treatments were conducted with the naturally occurring population. One trap per cage was used in the plot trials in 2020, as well as in 2021.

On field J, 24 (2 × 2 m) cages (12 control, 12 treatment) were set up. The sugar beet plants were spray treated with 1 × 10^13^ spores ha^−1^
*M. brunneum* Ma 43 as a dispersion formulation with 0.05% (*v*/*v*) NeoWett^®^ (wetting agent based on isotridecanol-polyglycolether, Kwizda Holding GmbH, Vienna, Austria) and 1.5% (*v*/*v*) PRESS^®^ Antischiuma (foam reducing agent based on silicone oil, Cifo Srl Bologna, Italy). Before treatment, 20 weevils were introduced into these plots. The same treatment was performed on the sugar beet plants of the open greenhouse experiment, but without the weevils.

### 2.3. Pest Monitoring

Pheromone traps were used to assess the infestation density of the weevil population. The traps consisted of a bucket (volume 3 L) and a piece of natural rubber mounted in the center with wire, to which an aggregation pheromone (Grandlure III&IV; Bedoukian Research Inc., Danbury, CT, USA) was added. Traps were buried in the soil (Appendix A) around the field borders with a distance of 10 to 15 m. The beetles are attracted by the pheromone, fall into the bucket and cannot escape due to the smooth walls. All traps were evaluated once a week, and the caught weevils were counted. Weevils from the plot trial were separated for further analysis (i.e., to determine the prevalence of weevil infection).

To evaluate whether *A. punctiventris* can be infected with the fungus by treated trap ditches, 300 weevils were collected from fields C and D shortly after they had entered the fields. Ten days after the spray application, weevils were manually collected from the trap ditch and up to 2 m in the field site and separated for quarantine. For comparison, additional control fields, which were untreated, were sampled. Two different approaches were performed: 180 weevils were manually caught from field H and individually kept for quarantine. For comparative sampling, pheromone traps were placed around the borders of field site I in the morning and all weevils were collected from the bucket in the afternoon, and then separated for quarantine. This should highlight whether the direct contact of weevils increases endemic mycosis. All weevils were kept at conditions optimal for fungal growth, 25 °C and 60% rH, and fed regularly with organic grown sugar beet seedlings. Upon death, cadavers were evaluated for fungal growth. Fungal isolates were taken from mycosed weevils and identified by morphology. *Metarhizium* spp. isolates were further prepared for molecular genetic identification.

At the Emigration field sites, the number of weevils leaving the field was assessed to determine whether there was selective directional migration of the beetles that could be attributed to for instance wind direction, position of the sun or neighboring sugar beet fields to better adjust future control measurements. For this purpose, the fields were closely bordered with buried pheromone traps (10 m distance between traps) and the weevils in the traps were counted weekly.

### 2.4. Evaluation of Metarhizium Abundance

Soil samples of all field sites were taken once before treatment and twice a year after application of the products with a core borer (for two years in a row). Samples were processed after Laengle et al. [20], plated onto selective Sabouraud-4%-Glucose agar media and were incubated for 14 days at 25 °C and 60% rH before counting colonies. Final abundance was calculated as a colony forming unit (CFU) per gram soil dry weight.

The abundance of *Metarhizium* on sugar beet leaves was determined by “washing” the spores off the leaves (10 leaves per plot were pooled). For this purpose, they were shaken for one hour in 0.1% (*w*/*v*) Tween 80 solution (0.6 to 10 cm^2^ mL^−1^), on a rotary shaker at 200 rpm in an Erlenmeyer flask. The solution was filtrated through a cotton plug which was inserted into a sterile 10 mL pipette (used upside down, [21]) to avoid leaf fragments in the spore suspension. From this suspension, 50 µL were plated in fourfold onto selective Sabouraud agar media. Incubation of plates was performed as mentioned above for the agar plates from soil samples. Abundance was calculated as CFU per square cm leaf surface.

### 2.5. Bioassay

For the bioassay (n = 4), sugar beet weevils were collected from an untreated field site and separated into 50 mL tubes containing entomopathogen free soil. After a 14-day quarantine, 18 weevils per treatment were chosen and put into 5 cm Petri dishes containing a moist paper towel. The weevils were directly inoculated with the spore suspension to ensure contact of host and pathogen. They were either treated with 50 µL 0.1% (*w*/*v*) Tween 80 solution (control) or with *M. brunneum* Ma 43, spore density of 5 × 10^6^ spores per weevil. The bioassay was incubated at 25 °C in a moist chamber, and the mortality of *A. punctiventris* was evaluated every day until all fungal treated weevils died.

### 2.6. Genetic Analysis of Fungal Isolates

To evaluate successful application of the product, isolates from the soil samples were identified with simple satellite repeat (SSR) PCR analysis. Therefore, from each sample two isolates belonging to the most common morphotype and one isolate with another morphology were selected. In addition, *Metarhizium* isolates from the mycosed sugar beet weevils, collected during the monitoring, were analyzed. The PCR was performed after Mayerhofer et al. [22] with the two marker sets I and V (Ma 2049, Ma2054, Ma2063, Ma195, Ma327, Ma2287). Species were assigned either based on comparison to already known genotypes from our previous work, or sequencing of the nuclear translation elongation factor-1α (EF1a) [23].

### 2.7. Data Analysis

Statistical analyses were performed with R version 4.1.0 (Free Software Foundation, Boston, MA, USA). The minimum spanning network (MSN) was created using the “poppr” package of R. Graphics were plotted with OriginPro 2020 (version 9.7.0.185; OriginLab Corporation, Northampton, MA, USA). SSR raw data were analyzed with GeneMarker (SoftGenetics, State College, PA, USA), sequencing data with BioEdit [24]. Phylogenetic tree was calculated with MEGA [25]. The bioassay was evaluated using Mathematica (Wolfram Research, Inc., Champaign, IL, USA) and a time- mortality analysis developed by Throne et al. [26], in which a formula is included to correct the treatment mortality with the control mortality.

## 3. Results

### 3.1. Bioassay

Direct treatment of the beetles with 5 × 10^6^
*M. brunneum* Ma43 spores per weevil and incubation at 25 °C resulted in a LT_50_ (lethal time) value of 5.5 days (±2.0) and a LT_90_ value of 10.2 days (±2.5) (n = 4, Figure 1). After death and further incubation, all weevils showed *Metarhizium* outgrowth, which was molecularly identified as the production strain.

### 3.2. Metarhizium Abundance and Ma 43 Identification in Soil

After the application of the granular product in the large-scale application field sites A and B, the number of the CFU in both fields changed significantly (Anova, *p* < 0.001, Figure 2A). The abundance of *Metarhizium* spp. increased from 376 ± 174 CFU per gram soil dry weight (field A) and 707 ± 114 CFU (field B) up to a maximum density of 5371 ± 935 and 20,338 ± 2755 CFU, respectively. The initial fungal density in the soil prior to treatment was significantly different compared to the densities in autumn 2020 and 2021. During winter, the density decreased by a factor of three to five. Nevertheless, a significant positive correlation (Pearson) between CFU and time could be observed for both field sites (r = 0.636, *p* = 0.011 and r = 0.552, *p* = 0.033, respectively; n = 15).

In field sites C and D, treated twice with the granular product as well as the additional spray application of the trap ditch, the *Metarhizium* abundance increased from 404 ± 63 and 592 ± 89 CFU, respectively, up to 4133 ± 360 and 6,458 ± 798 CFU per gram soil dry weight after the treatment (Figure 2B). Whereas the exclusive use of granules (fields A and B) led to an increased fungal abundance until autumn, the *Metarhizium* density in both fields with trap ditches initially increased in spring but decreased by half for field C and two-thirds for field D by autumn. No significant correlation between CFU and Time was observed for both field sites (*p* = 0.157 and *p* = 0.133 respectively; n = 12), although significant changes over time were observed (*p* = 0.009 and *p* < 0.001).

Field sites E, F and G were either treated before (E, F) or after (G) planting sugar beets in the experimental plots, with control plots in the same field site. Treated as well as untreated plots showed a low indigenous *Metarhizium* abundance before treatment (<210 ± 192 CFU). After a one-time application of the granular product, the density increased significantly (*p* < 0.001) up to 16,117 ± 3129 CFU per gram soil dry weight, whereas no significant changes were observed in the control plots (Figure 2C,D). In field site F, the control plots showed a higher indigenous *Metarhizium* abundance (779 versus 214 CFU) compared to the other control plots, however, after application, the fungal density only increased significantly up to 15,615 ± 2356 CFU (*p* < 0.001) in the treated field sites. In accordance with the above-mentioned plot trials, the fungal abundance in field site G also increased significantly after application from 70 ± 57 up to 14,897 ± 5555 CFU per gram soil dry weight (*p* < 0.001), whereas the density in the control field site did not change significantly. Due to soil cultivation in spring 2021, the soil from the plots got mixed, thus the fungal density in treated and untreated plots could no longer be distinguished. In field E, a low *Metarhizium* spp. abundance (<138 CFU) was found, whereas in field F and G high *Metarhizium* spp. abundance (>3552 CFU) was observed in both plot designs.

Both in the field J (2 × 2 m cages) and in the pot trial (open greenhouse), vital spores could still be isolated from the sugar beet leaves 40 days after spray application. In the field trial, the number of *Metarhizium* CFU cm^−2^ leaf surface differed significantly in both trial years (*p* < 0.001). Whereas the spray application in 2019 lead to about 270 CFU, almost 100 times this amount could be isolated directly after the spray application in 2020, with over 27,000 CFU. Even after 40 days, the values were still 50 times higher than the year before, while the quantity had decreased significantly (*p* < 0.001) in both years compared to the initial value (Figure 3A). A continuous decrease of *Metarhizium* density was also observed in the pot experiment (Figure 3B). However, by the end, significantly higher values were observed on treated plants compared to the control plants, on which *Metarhizium* could never be detected. After foliar application, an increase in the density of *Metarhizium* spp. was observed not only on the leaves but also in the examined soil samples. Whereas less than 320 (±172) CFU g^−1^ soil dry weight were determined in all control samples as well as in the samples evaluated before application, the density increased up to 18,860 (±2790) CFU in field J and 10,580 (±1900) CFU in the pot trial.

Before application of GranMet^TM^-P, the production strain *M. brunneum* Ma 43 was not identified among the isolates investigated prior to treatment (indigenous isolates). The main part (94.3%) of indigenous *Metarhizium* isolates belonged to the species *M. robertsii* (J.F. Bisch., Rehner & Humber). After the treatment, *M. brunneum* Ma 43 became the dominant (53.8%) species/isolate of the isolated *Metarhizium* spp. (Figure 4, allele sizes available in Appendix A). As already suspected from the obtained abundance data, tillage in spring of the field sites E, F and G led to a mixing of the soil from treated and untreated plots–thereafter Ma 43 was also present in untreated plots.

### 3.3. Pest Monitoring and Mycosis

Emigration of *A. punctiventris* in the area was evaluated from the trapped weevils of the Emigration field sites (pheromone traps around the fields with 10 m between traps). The mean value of emerging weevils was significantly different between the field sites (Anova, *p* < 0.001, F- value = 397.74, dF= 12; n = 1381 from 13 fields). The lowest number of evaluated weevils was 797 individuals caught over a period of 11 weeks from one field (mean value of 15 ± 8 weevils per trap). The highest number was 267,594 weevils caught in total at one site (mean value: 2549 ±1028 per trap). Due to this high variation, the number of weevils per trap was ranked for all fields, to plot the number of weevils against the compass direction of the trap location (Figure 5). Although no significant preference of emigrating could be evaluated, a tendency of a higher number of weevils could be detected SSW and WNW of the fields. Lowest weevil numbers were found in the eastern side.

Field sites E and F, both evaluated the year after sugar beet cultivation, showed different numbers of emigrating weevils. Whereas in field site E a total number of 3447 weevils were caught in the 12 plots, only 301 were caught in field F, which is about a tenfold lower number of weevils. Interestingly, also in 2021, the second year without sugar beet cultivation, emerging weevils could be found in the plots. Nearly 3% of the initially first emerging adults were found (2.73 and 2.99%, respectively). Similar to the emigration field sites, a varying number of indigenous weevils could be observed in fields E and F.

The different sampling techniques for evaluating the prevalence of fungal infection of the weevils showed that it is important to collect the weevils individually from the field if they are needed alive for further testing. When the adults were trapped massively with the pheromone traps and kept at conditions optimal for fungal growth, almost half of the weevils (e.g., 49.5% in field H) were mycosed within one month (n ≥ 300), whereas the individually sampled weevils from field site I only showed mycosis of 3.3% (n ≥ 180). Weevils collected shortly after having migrated through the treated trap ditches of field sites C and D showed a mycosis of 16.6% and 19.6%, respectively. Weevils from field site E and F showed different infection rates, whereas in field side E more than 25% mycosed, field side F showed only 11% mycosed weevils. The continuously treated field site B, treated twice before growing sugar beets, showed increased mycosis in emerging weevils the year after the cultivation. During quarantine, 70% of the collected weevils mycosed (n ≥ 180 per site). Most of the infections were caused by *Beauveria* spp. with at least 44% of mycosed weevils collected from the treated field sites being infected by this fungus (Figure 6). From the weevils immigrating into field sites C and D across the trap ditch, more than 50% (52.6 and 62.5, respectively) of the *Metarhizium* mycosed individuals were infected by the production strain Ma 43. Weevils from field site B, that underwent their development in the treated soil and then started to emigrate, were predominantly infected by the production strain, with 88.9% of all *Metarhizium* isolates being identified as *M. brunneum* Ma 43. Fourteen percent of the weevils collected from field J 18 h after the spray treatment with GranMet^TM^-WP were already infected with *Metarhizium*.

Overall, aside from the production strain, three different species, namely *M. lepidiotae* (Driver & Milner), *M. robertsii* and *M. brunneum*, were isolated from the *Metarhizium* mycosed weevils (allele sizes available in Appendix A). The predominant species with the greatest diversity of multilocus genotypes (MLG) was *M. robertsii*, which was also isolated most frequently from the untreated soil samples. Excluding the production strain (*M. brunneum* Ma43), MLG 3 was the genotype isolated most frequently from mycosed weevils as well as from the soil of all trial sites (Table 2).

## 4. Discussion

Apart from the use of nematodes [27,28], which were only used against the larvae of the sugar beet weevil *A. punctiventris* in the soil, this large-scale study is one of the first in almost a hundred years (except a few studies using *Beauveria bassiana* [14,15]) to test the use of a biologically based insecticide against the weevil pest. We aimed to test the effectiveness of a preventive use of the entomopathogenic fungus *Metarhizium brunneum* against the weevil in all stages of its development. To directly control the adult beetles, a foliar spray application containing the same fungus was carried out (field J). In addition, a spray treatment of the trap ditches with *M. brunneum* was performed when the adult beetles were immigrating into the sugar beet field (field C and D).

For the spray treatment trials, the spore viability of the powder product GranMet^TM^-WP was tested, and accordingly a maximum of 4.4 kg per hectare was used. Although this dosage was lower than that mentioned in older literature, our production strain *M. brunneum* Ma 43 could nevertheless be established in the treated areas over a period of several months and remain at a fungal density of >1000 CFU g^−1^ soil dry weight. This persistence was above the maximum background level defined by Scheepmaker et al. [29]. The conclusion that *Metarhizium* spp. seems to be suitable for in situ use in sugar beet cultivation is further supported by the fact that more than a hundred years ago, the fungus was already shown to successfully infect and kill the sugar beet weevil in a sugar-beet field [9]. However, a direct comparison of older studies from the nineteenth and twentieth centuries on the use of *Metarhizium* spp. against the sugar beet weevil is only useful and possible to a limited extent, due to the small number of conducted trials as well as the lack of direct pathological findings. For example, Krassilstchik [7] achieved a 55–80% reduction in beetle density after about 2 weeks with a spore application of 8 kg per hectare, but there is no information on the actual number of vital spores the author used per hectare. Furthermore, abundance data for this application was not published either, so that we can only speculate about the actual spore density and its persistence. In addition, our abundance values of more than 1000 CFU g^−1^ soil dry weight after the application of the products were in contradiction with the results published by Pospelov [30], who performed similar experiments. The author could only prove *Metarhizium* infections in the trap ditches, but no accumulation of the fungus in the soil, nor an effect of the fungus on adult beetles when applied directly in the field. The reasons for this cannot be determined, because no information on the used spore concentrations, the application methods, the quality of the spore product nor the product formulation could be gained from Pospelov’s report [30].

In their studies, Krassilchik [7], Ivanov [31] and Pospelov [30] observed mycosis rates of 20–80% and 55–80%, respectively. The authors indirectly attributed the strong deviations in the abundance values to the pH value of the soil in the experimental fields: Pyatnitzkiï [32] reported that *Metarhizium* successfully infects insects, especially in more acidic soils. Pospelov [30] advocated the use of *Metarhizium*, especially in the area around Kiev, because he found that the “white muscardine”, i.e., the infestation with the insect-killing fungus of the genus *Beauveria* spp. was not successful in this region. The soil pH around Kiev is slightly acidic, which would also confirm the authors’ theory that acidic soils can improve the persistence and development of *Metarhizium*. However, the sites investigated in our study were characterized by neutral to alkaline soils [33]. Nevertheless, our abundance values showed a maximum density of nearly 20,000 CFU g^−1^ soil dry weight, which were directly attributable to the presence of our applied production strain *M. brunneum* Ma 43. Interestingly, we also found a high indigenous density of *Beauveria* spp. in our soil samples. In this study, we focused on *Metarhizium* spp. density and genetic composition, but analyzing the *Beauveria* spp. population composition and virulence might be addressed in further studies. Previous experiments have indicated that the abundance of applied biological control agent (BCA) strains is promoted by pest infestation and application frequency of the active ingredient. Abiotic and edaphic factors are principally important test criteria but are usually of secondary importance, and their influence on the fungus is often not yet well understood [34,35]. Unfortunately, due to the very inhomogeneous distribution of pest infestation, we could not observe a direct effect on the pest density. In our first control approaches, the weevils were already present or were just about to infest newly planted sugar beet fields. As such, the fungus only had a short time to infect the host before the plants were damaged. To prolong the possible infection time frame and increase efficacy, one should consider applying the fungus in spring, before the start of migration, on fields where sugar beet was grown in the previous year. Furthermore, weevils emigrating from these fields should be trapped in treated trap ditches.

For successful biological control, the importance of knowledge of the biology of *Asproparthenis* should not be neglected. We were able to show that the weevils not only hatch and emerge in the year following sugar beet planting, but also two years after the first infestation as a migratory beetle: even in the second year of crop change, about 3 percent of the beetles emerged from the soil of the experimental plots. Regarding a sustainable control of the *Asproparthenis* population, this observation is of utmost importance, because despite a supposedly low number of weevils, the population of the beetle in the region can increase significantly. Assuming that each female lays between 80 and 100 eggs on average [11,36], the damage threshold of the weevil in the cultivation regions (i.e., damage threshold for sugar beet weevils is 0.1 insects m-^2^ during emergence of beet seedlings, [37]) can be reached again by the surviving weevils within the next generation (considering optimal conditions for egg weevil development). This active emigration of adults from old sugar beet fields over several years was also described by Pyatnitzkiï [32,38] and could also explain the increase in population density in the cultivation region in Lower Austria despite crop change (annual crop change with a four-year gap between two sugar beet plantings). In contrast, Tielecke [11] did not observe any weevils that hibernated for two years, but attributed this to the low overwintering depth of the pest population (75% at 10 cm depth). This is in accordance with the findings of Pyatnitzkiï [32], who was able to establish a connection between the overwintering depth and the hibernation length of the beetles (dormant weevils in depths of 40–50 cm). However, while Pyatnitzkiï’s studies reported a relatively high proportion of dormant beetles (up to 20% for two years, up to 10% for three years; [38]), in the second year the proportion of emerging beetles in our experimental plots was six times lower (overwintering depth 10–50 cm).

The direction of the beetles’ migration is also of importance for a pest prognosis and, consequently, for preventive control measures. For this reason, the migration of the weevil from thirteen different field sites across the entire sugar beet cultivation area in Lower Austria was investigated. Varying observations from previous investigations (e.g., no direction of migration, migration according to location of sun, migration according to soil grooves) do not allow a clear conclusion to be drawn [10,11,32]. Although at present no significant cardinal direction of emigration could be proven in this study either, a tendency to migrate towards the south and west could be detected. For further studies it is necessary to pay attention to other environmental factors (including location of nearest beet field, vegetation surrounding the field, wind direction, other topological obstacles) as well as their combined effect. If the direction of emigration can be predicted, this will enable a more targeted *Metarhizium* treatment. Similarly, to an application technique mentioned by Hunter [39], barrier strips in the migration direction of the weevil can be established, thus increasing the possibility of a successful fungal infection and leading to a higher mortality of the emigrating weevils.

This first large-scale field trial will serve as the promising basis for further research into the right combination of IPM measures and additional studies of the biology of the weevil. We also consider it important to apply the fungus repeatedly as well as before the weevil starts emigrating. This increases the likelihood of controlling all development stages of the weevil. Sugar beet fields which were cultivated in the previous year should be treated in spring shortly before the start of migration, in combination with other IPM measures (e.g., also in combination with trap ditches). Initial spray treatments of the trap-ditches have been effective, and are now being investigated in more detail for sustainable barrier system support as part of a follow-up project.

## Figures and Tables

**Figure 1 pathogens-12-00099-f001:**
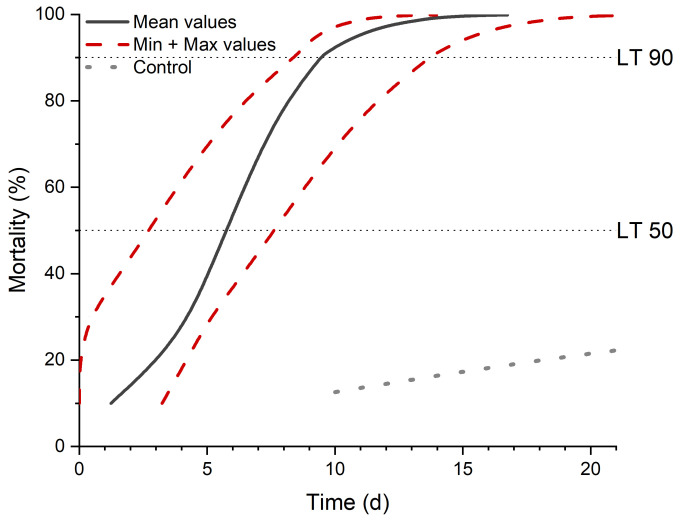
Time-Mortality analysis of *A. punctiventris* weevils inoculated with the fungal bioinsecticide. Mean values (black line) and minimum and maximum values (red lines) are shown. Dotted lines indicate LT_50_ and LT_90_ values. Results are obtained by a probit analysis corrected by control group values [26].

**Figure 2 pathogens-12-00099-f002:**
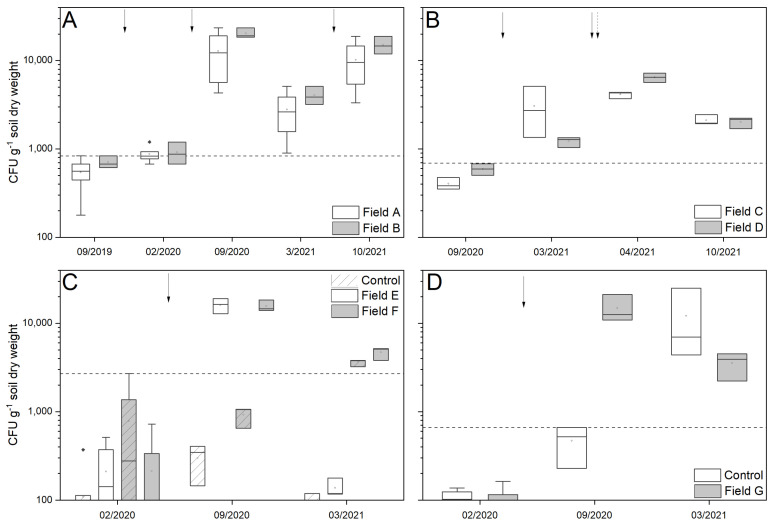
Development of *Metarhizium* spp. abundance in soil in all trial sites. (**A**) continuous granular application, (**B**) granular application with spray application of the trap ditch, (**C**) application after sugar beet cultivation, (**D**) application before sugar beet cultivation. Arrows indicate application time (dashed arrow indicates the spray application), and dashed line shows maximum indigenous density without soil disturbance.

**Figure 3 pathogens-12-00099-f003:**
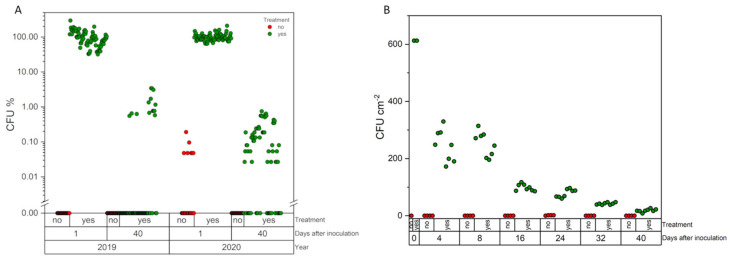
*Metarhizium* spp. CFU isolated from leaves after spray treatment. (**A**) CFU in percent calculated from mean value of day 0 in the field, (**B**) CFU per cm^2^ leaf surface in the pot trials. Each dot represents an individual value of a single plate.

**Figure 4 pathogens-12-00099-f004:**
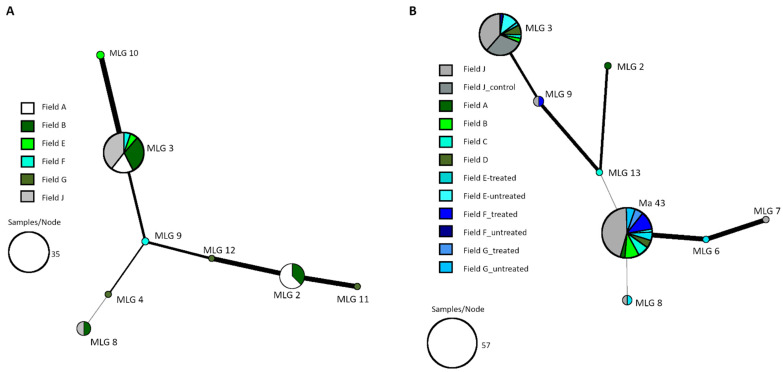
Minimum spanning network (MSN) showing the relationship between the SSR genotypes isolated from the soil of the field sites before (**A**) and after (**B**) spray application of *M. brunneum* Ma 43. Circle sizes are proportional to the number of isolates belonging to one MLG, the thickness of the line is proportional to genetic SSR-based similarity of genotypes.

**Figure 5 pathogens-12-00099-f005:**
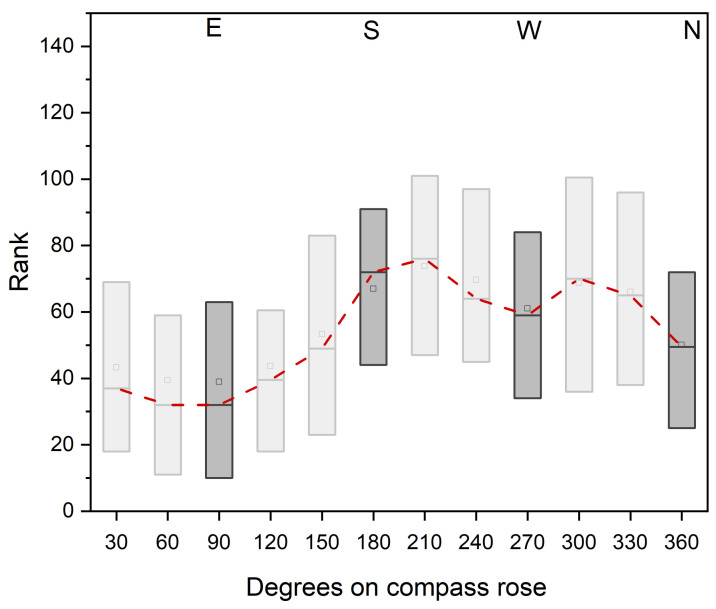
*A. punctiventris* beetles caught with pheromone traps, which were installed at regular intervals of 10 m along the edges of the fields. Thirteen trial sites with an area size between 2.4 ha and 13.1 ha were sampled throughout Lower Austria in the planting season 2021. Ranked number of weevils per location is plotted against the location in which they were caught in degrees on a compass rose (360°; E = East, S = South, W = West and N = North).

**Figure 6 pathogens-12-00099-f006:**
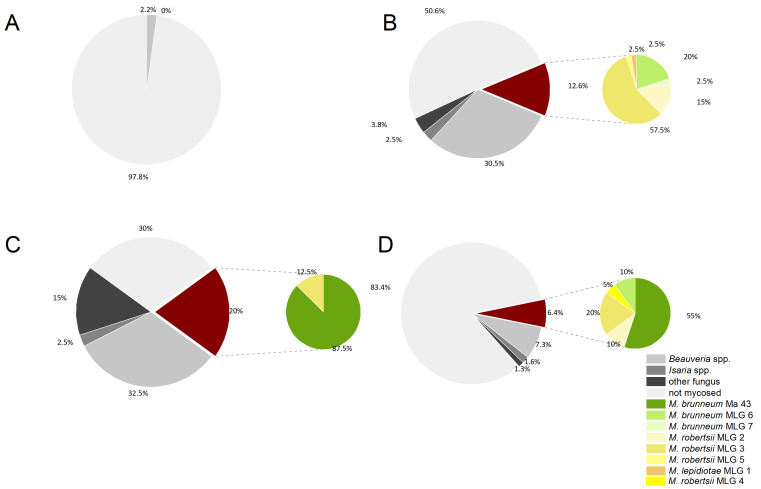
Mycosis evaluation of sugar beet weevils from (**A**) control sampled separately, (**B**) control sampled from pheromone traps, (**C**) field site B (emerging beetles after two-time application) and (**D**) field site C (immigrating beetles that had just crossed the sprayed trap ditch). The larger grey/red circle indicates percentage of mycosis with different fungi, and the smaller green/yellow circle shows percentage of different *Metarhizium* species isolated from weevils.

**Table 1 pathogens-12-00099-t001:** Specification of field sites in Lower Austria treated with the fungus *M. brunneum*.

Location	Coordinates	Field/Plot Size	Treatment	Sample Taken	Sugar Beet
Field A	48.36689° N/16.04227° E	0.8 ha	3 × Granular	Soil	2020
Field B	48.37026° N/16.34470° E	0.8 ha	3 × Granular	Soil	2020
Field C	48.43198° N/16.03101° E	0.7 ha	2 × Granular; 1 × Liquid	Soil/Weevil	2021
Field D	48.43504° N/16.03315° E	0.7 ha	2 × Granular; 1 × Liquid	Soil/Weevil	2021
Field E	48.64448° N/16.02301° E	0.6 ha/4 m^2^	Granular Plot application	Soil/Weevil	2019
Field F	48.66603° N/16.01579° E	0.6 ha/4 m^2^	Granular Plot application	Soil/Weevil	2019
Field G	48.54607° N/16.07112° E	0.6 ha/4 m^2^	Granular Plot application	Soil/Weevil	2020
Field H	48.46394° N/15.92893° E	6.4 ha	No treatment	Weevil	2020
Field I	48.37637° N/15.95542° E	5.1 ha	No treatment	Weevil	2021
Field J	48.32155° N/16.03788° E	4 m^2^	Liquid (leaf application)	Soil/Leaf/Weevil	2019/20

**Table 2 pathogens-12-00099-t002:** Total number of multilocus genotypes (MLG) of *Metarhizium* species, without the application strain, isolated from mycosed weevils and number of the MLGs in collected soil samples from the trial sites (n = 7) in Lower Austria.

MLG	Species	N° Found on Weevils	N° Found in Soil	N° of Trial Site
1	*M. lepidiotae*	1	0	1
2	*M. robertsii*	14	14	2
3	*M. robertsii*	52	33	5
4	*M. robertsii*	1	1	1
5	*M. robertsii*	1	0	1
6	*M. brunneum*	12	1	2
7	*M. brunneum*	1	0	1

## Data Availability

All additional data can be obtained from the corresponding author upon request.

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
