# Peer review of "Integrated Biological Control of the Sugar Beet Weevil Asproparthenis punctiventris with the Fungus Metarhizium brunneum: New Application Approaches"

_pathogens, 2023, doi:10.3390/pathogens12010099_

Round 1
Reviewer 1 Report
I am concerned with the level of replication in this study. In figure 1 the two sites per treatment are shown. Just by chance if one field acts differently from another. Three replicates is the accepted minimum standard level of replication for such studies. The number of trials done and different variations of a general question whether the application of the biocontrol agent results in higher CFUs. Fewer tests but more replication of a test would have been preferable.
The study only cites 30 references and you have quite a few very old references that is commendable that you searched for all historic work on your crop pest system. However, the current literature specifically needs to be consulted and the current best IPM practices introduced. Papers that are more than 100 years old can contain valuable information but the field would have progressed much in a century. Consider that other literature on other weevil pest-crop systems using Metarhizium as a biocontrol agent will be relevant. Your literature review and discussion is not making sufficient use of the current scientific literature.
In the abstract the following statement is made: “In conclusion, the various Metarhizium application strategies, which are already available or in testing, must be implemented to ensure control in both old and new sugar beet fields.”
The first six lines of the introduction has no references. Here you should cite important papers in the field not necessarily about your pest-crop system.
Table 1. Details on fields H and I need to be added. It is currently difficult to follow your study design. Also, why is it important to list the one site specifically as the AGRANA Research field site? Do you expect this field site to differ from other field sites? Please provide more details on how sites were selected.
The purpose of the sampled emigration fields is not well explained and are also not well detailed in the supplementary apart from their position. The rationale and design should be described in the main article.
Lines 129-130: “Just before weevil immigration started, 129 the field sites were surrounded by a trap ditch (12 cm width and 40 cm depth) produced 130 with a cable trencher (Cosmeco V1).” How do you know that weevil immigration had not started? Actually there is no explanation on this process in crop fields and how you measured this relevant to your sampling design.
Lines 153-154: were bucket pheromone traps buried in the soil? How do the weevils enter the traps, can they fly? Essential detail is missing.
Line 162: replace “:” with “.”
Lines 164 to 167: “Two different approaches were performed: firstly, 180 weevils were selected in one field (further referred to as field site H) the same way as in the treaded field sites and additionally, pheromone traps were placed around another field site (field site I), emptied in the morning and collected in the afternoon to quarantine the freshly caught weevils.” Despite several reads I cannot follow this text, please state more clearly. Change “treaded” to ‘treated’.
Line 193: “2.5. Bioassay” – The replication with this section is not clear, how many replicates were there?
Figure 6: You have to add a mortality line of the control group to this figure. The text, “Results are obtained by a probit analysis corrected by control group values.” does not provide any reference for using this approach and whether it is appropriate to do so.
Lines 241 to 243: “Nevertheless, a significant positive correlation (Pearson) between CFU and time could be observed for both field sites (r=0.636, p=0.011 and r=0.552, p=0.033, respectively).” Please add the sample size for both correlations.
Lines 250-252: Same as above.
Figure 2: why is there only on red dot where the “no” treatment is applied (for day 1 in A and day 0 in B)?
Lines 309-311: “Emigration of A. punctiventris in the area was evaluated from the Emigration field sites. The mean value of emerging weevils was significantly different between the field sites (Anova, p<0.001).” This experiment is not sufficiently explained, i.e., where are the emerging weevils being collected from, the centre of the site or the edge. Impossible to interpret these results. Also the Anova result reported is not sufficient at present: what was the F-statistic and the sample size. One cannot just provide a p-value.
Author Response
Dear Reviewer,
Thank you for taking the time to improve our manuscript with your feedback.
Here is a point-by-point response to your comments and concerns. We have highlighted the changes within the manuscript:
I am concerned with the level of replication in this study. In figure 1 the two sites per treatment are shown. Just by chance if one field acts differently from another. Three replicates is the accepted minimum standard level of replication for such studies. The number of trials done and different variations of a general question whether the application of the biocontrol agent results in higher CFUs. Fewer tests but more replication of a test would have been preferable.
We agree that more repetitions are certainly beneficial. The particularity of our work is that we were able to test different application strategies of Metarhizium brunneum at sites with a naturally heavy infestation, over several planting seasons. We were careful to ensure that at least one replication of each measure was performed over two seasons. Originally, at least three replicates were carried out for each trial, but areas had to be excluded from the evaluation because some fields were given up due to crop loss or because the agricultural set up was changed without our agreement. In addition, it must be considered that in the plot trials, 6 repetitions per treatment were carried out.
The study only cites 30 references and you have quite a few very old references that is commendable that you searched for all historic work on your crop pest system. However, the current literature specifically needs to be consulted and the current best IPM practices introduced. Papers that are more than 100 years old can contain valuable information but the field would have progressed much in a century. Consider that other literature on other weevil pest-crop systems using Metarhizium as a biocontrol agent will be relevant. Your literature review and discussion is not making sufficient use of the current scientific literature
The argument of the number of citations does not seem necessary, as all important papers on this pest, which are also relevant to the paper, have been cited. With the review paper [12], the most current and also practically relevant IPM techniques for A. punctiventris control were presented. Someone has to accept, chemical control strategies are still favoured over all other possibilities I Central Europe. For this reason, no more actual papers were published, with the exception of those full papers reviews already referred in our manuscript.
Nevertheless, it is also well known that entomopathogenic macro- and micro-organisms are used worldwide to control weevil pests. But this was not the aim of our work, we wanted to focus on this very special sugar beet weevil in particular, which has been causing great damage for a long time without any new solutions being found to control it. Until now, only chemical insecticides are referred as a practice-relevant control measure. [12]
In the abstract the following statement is made: “In conclusion, the various Metarhizium application strategies, which are already available or in testing, must be implemented to ensure control in both old and new sugar beet fields.”
The citation of the statement is correct, but we don't see a connection to a question?
The first six lines of the introduction has no references. Here you should cite important papers in the field not necessarily about your pest-crop system.
References werea added: “Since the beginnings of agriculture, there has been a struggle to avoid pest populations in order to guarantee our food supply [1, 2]. The more abundant, severe and crop-threatening a pest is, the more effort is needed to solve the pest problem [3]. Over time, however, environmental conditions can change, and some pests may become more frequent or have higher population densities [4], as is the case for the sugar beet weevil (Asproparthenis punctiventris Germar, previously Bothynoderes punctiventris Germar, Coleoptera: Curculionidae) [5,6]”.
- IPPC Secretariat, Guillino, M.L.; Albajes, R.; Al-Jboory, I.; Angelottii, F.; Chakraborty, S.; Garret, K.A.; Hurley, B.P.; Juroszek, P.; Makkouk, K.; Pan, X.; Stephenson, T. Scientific review of the impact of climate change on plant pests – A global challenge to prevent and mitigate plant pest risks in agriculture, forestry and ecosystems. FAO on behalf of the IPPC Secretariat, Rome, Italy, 2021; 1-72.
- Alexandratos, N.; Bruinsma, J. World agriculture towards 2030/2050: the 2012 revision. FAO, ESA Working paper, Rome, Italy, 2012; 1-147.
- Gvozdenac, S.; Dedić, B.; Mikić, S.; Ovuka, J.; Miladinović, D. Impact of Climate Change on Integrated Pest Management Strategies. In Climate Change and Agriculture, Benkeblia, N., Ed.; John Wiley & Sons Ltd.: New Jersey, USA, 2022, 311-372.
- Andrew, N. R.; Hill, S.J. Effect of Climate Change on Insect Pest Management. In: Environmental Pest Management, Moshe Col, M.; Wajnberg, E., Eds. John Wiley & Sons Ltd.: New Jersey, USA, 2017, 195–223.
- Drmić, Z.; Čačija, M.; Virić Gašparić, H.; Lemić, D.; Bažok, R. Phenology of the sugar beet weevil, Bothynoderes punctiventris Germar (Coleoptera: Curculionidae), in Croatia. Entomol. Res. 2018, 109 (4), 1–10.
- Klukowski, Z.; Piszczek, J. Distribution of damages in Poland caused by the sugar beet weevil (Asproparthenis punctiventris Coleoptera: Curculionidae). J. Plant Prot. Res. 2021, 61, 311–313.
Table 1. Details on fields H and I need to be added. It is currently difficult to follow your study design. Also, why is it important to list the one site specifically as the AGRANA Research field site? Do you expect this field site to differ from other field sites? Please provide more details on how sites were selected.
Fields H and I are added. The research field was special because it is a field that was cultivated specifically for research purposes and is not managed by the farmer. However, we have changed the name of the field to avoid misunderstandings. The selection criteria for the fields have been added.
The purpose of the sampled emigration fields is not well explained and are also not well detailed in the supplementary apart from their position. The rationale and design should be described in the main article.
The purpose of the emigration fields was explained in section 2.3, last paragraph. We have now added more information to better describe this trial, see also 2.1 Trial sites: “…… 13 field sites (further referred to as “Emigration fields”, Table S1), were selected to evaluate the emigration of weevils (2.3). The fields were chosen based on the results obtained from autumn monitoring of overwintering weevils the previous year, which predicted a high number of migrating weevils. No application was performed on the field sites.”
Lines 129-130: “Just before weevil immigration started, 129 the field sites were surrounded by a trap ditch (12 cm width and 40 cm depth) produced 130 with a cable trencher (Cosmeco V1).” How do you know that weevil immigration had not started? Actually there is no explanation on this process in crop fields and how you measured this relevant to your sampling design.
Weevil migration was regularly monitored by the authors and farmers. Furthermore, the current weather situation was taken into account, as it influences the behaviour of the weevils and the migration time. In addition, excavations were carried out and the soil depth at which the weevils were located was determined. We included this information in the manuscript.
Lines 153-154: were bucket pheromone traps buried in the soil? How do the weevils enter the traps, can they fly? Essential detail is missing.
Thank you, we missed explaining how to set the traps. “The traps were buried in the soil ....”. The beetles are attracted by the pheromone, fall into the bucket and cannot leave it because of the smooth walls. This information was included in the text. An additional picture of the buried trap was added to the supplementary data file.
Line 162: replace “:” with “.”
Punctuation was changed according to suggestion.
Lines 164 to 167: “Two different approaches were performed: firstly, 180 weevils were selected in one field (further referred to as field site H) the same way as in the treaded field sites and additionally, pheromone traps were placed around another field site (field site I), emptied in the morning and collected in the afternoon to quarantine the freshly caught weevils.” Despite several reads I cannot follow this text, please state more clearly. Change “treaded” to ‘treated’.
We have reworded this paragraph.
Line 193: “2.5. Bioassay” – The replication with this section is not clear, how many replicates were there?
The bioassay was repeated 4 times. N=4 was transferred from the end of the paragraph to the first sentence for better understanding.
Figure 6: You have to add a mortality line of the control group to this figure. The text, “Results are obtained by a probit analysis corrected by control group values.” does not provide any reference for using this approach and whether it is appropriate to do so.
The bioassay was carried out according to standard mortality analysis and the data was calculated according to Throne et al [17] in a time mortality analysis using Mathematica, as mentioned in 2.7 Data analysis. This reference was omitted below the image but has been inserted. Since the data from the treatment was calculated against the data from the control for this evaluation, the results were already corrected with the deaths in the control. For the presentation of the control, the control data must be calculated against itself, which did not seem to make much sense. However, since this will be a more frequent issue, we have now also calculated this data and inserted it into the graph.
Lines 241 to 243: “Nevertheless, a significant positive correlation (Pearson) between CFU and time could be observed for both field sites (r=0.636, p=0.011 and r=0.552, p=0.033, respectively).” Please add the sample size for both correlations.
Lines 250-252: Same as above.
Sample sizes were added in both cases.
Figure 2: why is there only on red dot where the “no” treatment is applied (for day 1 in A and day 0 in B)?
Due to the logarithmic scale of the graph, zero values are not visible. We changed the scaling for a better presentation of the results.
Lines 309-311: “Emigration of A. punctiventris in the area was evaluated from the Emigration field sites. The mean value of emerging weevils was significantly different between the field sites (Anova, p<0.001).” This experiment is not sufficiently explained, i.e., where are the emerging weevils being collected from, the centre of the site or the edge. Impossible to interpret these results. Also the Anova result reported is not sufficient at present: what was the F-statistic and the sample size. One cannot just provide a p-value.
The weevils were collected in the pheromone traps surrounding the field sites. This is mentioned in 2.2, but a small comment was made in the results to clarify the sampling. Statistics were added.
Reviewer 2 Report
This manuscript reports the evaluation of field efficacy of an entomopathogenic fungus on an insect pest whose distribution has expanded and damage has increased in the last two decades. Each field assay type was realized with few replicates but there are different experiments and all together can answer to the questions proposed by the authors.
However, there are some aspects to correct /improve in the manuscript (in order of the appearance in the manuscript), especially on the M&M section.
More important issues
- Keywords: we should not repeat words of the title.
- Introduction:
It is not exactly correct to say that “Biological control of the sugar beet weevil with entomopathogenic fungi has been neglected for the last 100 years”. Please see the studies with Beauveria bassiana:
- Beratlief, Z. (1979). Investigations on the entomopathogenic fungus Beauveria bassiana (Bals.) Vuill. and its action on the Colorado beetle (Leptinotarsa decemlineata Say) and the beet weevil (Bothynoderes punctiventris Germ.). Analele Institutului de Cercetari pentru Protectia Plantelor, 15, 233-241 – reporting good field efficacy (about 70% mortality);
- Slavchev, A. (1988). Possibilities for reducing the numbers of Bothynoderes punctiventris Germ. (Coleoptera, Curculionidae) by the entomopathogenic fungus Beauveria bassiana (Bals.) Vuill. Rasteniev" dni Nauki, 25(6), 97-101 – reporting about 90% mortality with soil insecticides that are not currently registered to use in EU.
I think that the order of your objectives should be changed. It seems strange to have one on the biology in the middle of two about the efficacy evaluation of commercial products and other about evaluation of natural occurring infection.
Homoptera (line 55) is not considered as order or suborder anymore. Please replace by “Hemiptera: Sternorryncha”
- Material and methods
The description of the several assays are a little bit confusing. Can the authors provide a graph, an image to support this description, for example in the Supplementary material? A photo of the trap ditch should be also included. And please clarify what treatments were made on plots C and D: on line 375 we mention a foliar application that was not mentioned in M&M section.
Lines 103 – 108 – The soil used in the pots experiment was Metarhizium free. How do you know that? And about other entomopathogens that could interfere in the experiment (e.g. Beauveria bassiana)? Was it sterilized?
Lines 146-151 - On the ANAGRA field the plants were sprayed with a formulation of (I suppose) the same strain of M. brunneum but nothing is told about it. And what are NeoWett® and PRESS®? Provide the name of the substance in both commercial products.
- Results and discussion sections
Figure 6 – the first figure included in the text is figure number 6?? Please renumber all the figures
Lines 247-249 – I disagree. The first treatment on fields A and B was done in 2019 in Autumn; on fields C and D the first treatment was done in Autumn 2020. These are the ones that can be compared because they were done at the same season and in all these cases there was an increase in fungus level.
Line 344-345 – Can you discuss this finding?
Lines 347-358 – Strain designations in the text and in Figure 5 are not the same. Please correct the names of the strains in the figure (M. brunneum Ma43 and MLG3 are not in the figure).
Minor issues
Throughout the text put a space between the value and symbol of the unit (e.g. 25 °C, 25 m)
Line 166 – “Treaded”? Should it be “treated”?
Line 196 – Petri dish (initial capital letter)
Lines 211-212 – Reference [16] should be on the end of line 211
Figure 2A – please consider the Y-axis beginning at 0.0 and not 0.1% CFU. If not, there will be no red circles in three of the four cases and we can think that no determinations were made in control pots.
Line 332 – delete the “-“ between “The” and “difference”
Line 354 – The authority of M. robertsii should be provided on line 297 (the first time it is mentioned)
Line 358 – What is the production strain (M. brunneum Ma43?)
Line 384 – Metarhizium brunneum or Metarhizium spp.?
Author Response
Dear Reviewer,
Thank you for taking the time to improve our manuscript with your detailed feedback and suggestions.
Here is a point-by-point response to your comments and concerns. We have highlighted the changes within the manuscript.
Keywords: we should not repeat words of the title
New keywords have been adjusted.
It is not exactly correct to say that “Biological control of the sugar beet weevil with entomopathogenic fungi has been neglected for the last 100 years”. Please see the studies with Beauveria bassiana:
Thank you for the citations, we have included them in our introduction: “Biological control of the sugar beet weevil with entomopathogenic fungi has been neglected for the last 100 years. Only a few studies have reported the use of Beauveria bassiana against the sugar beet weevil [14,15], Metarhizium spp., however, was hardly mentioned”.
For your better understanding: We focused on Metarhizium in our work, so we did not elaborate on the potential of Beauveria. The successful use of Beauveria is reported mainly in arable farming in North America, while in Europe the focus has been on the use of Metarhizium. In the future, of course, all alternatives should be considered.
I think that the order of your objectives should be changed. It seems strange to have one on the biology in the middle of two about the efficacy evaluation of commercial products and other about evaluation of natural occurring infection.
We changed the order of the objectives in accordance with the reviewer’s suggestion.
Homoptera (line 55) is not considered as order or suborder anymore. Please replace by “Hemiptera: Sternorryncha”
Homoptera was replaced following the reviewer’s suggestion
The description of the several assays are a little bit confusing. Can the authors provide a graph, an image to support this description, for example in the Supplementary material? A photo of the trap ditch should be also included. And please clarify what treatments were made on plots C and D: on line 375 we mention a foliar application that was not mentioned in M&M section.
We have already adjusted point 2.1 according to Reviewer 1 and hope to have eliminated the ambiguities. A picture of the trap ditch has been added to the additional data (Figure S2). To better assign the treatments in the discussion, the names of the fields have been included. A foliar spray application is not equivalent to the spray treatment, which refers to the treatment of the trap ditch.
Lines 103 – 108 – The soil used in the pots experiment was Metarhizium free. How do you know that? And about other entomopathogens that could interfere in the experiment (e.g. Beauveria bassiana)? Was it sterilized?
The soil was tested for the presence of entomopathogenic fungi before use after Laengle et al. using selective Sabouraud agar media. This information is now included in the text.
Lines 146-151 - On the AGRANA field the plants were sprayed with a formulation of (I suppose) the same strain of M. brunneum but nothing is told about it. And what are NeoWett® and PRESS®? Provide the name of the substance in both commercial products.
Information on NeoWett and PRESS as well as the name of the used strain were added.
Figure 6 – the first figure included in the text is figure number 6?? Please renumber all the figures.
Thank you for the information, the renumbering was forgotten after restructuring of text. Numbers were adjusted.
Lines 247-249 – I disagree. The first treatment on fields A and B was done in 2019 in Autumn; on fields C and D the first treatment was done in Autumn 2020. These are the ones that can be compared because they were done at the same season and in all these cases there was an increase in fungus level.
The notice was correct, all densities increased after application. The text was specified to make the statement clear.
Line 344-345 – Can you discuss this finding?
We agree, that the Beauveria findings are also very interesting. In our first draft we discussed including more data on our Beauveria results. However, we decided that going into too much detail about Beauveria would lead to the loss of focus of this paper. This paper is not about Beauveria, and therefore we focused only on Metarhizium and the well known production strain Ma 43 we applied. We now included a short sentence predicting further results on this topic, because we want to look into this field more closely in the future. See also line: 442-445.
Lines 347-358 – Strain designations in the text and in Figure 5 are not the same. Please correct the names of the strains in the figure (M. brunneum Ma43 and MLG3 are not in the figure).
Thank you, we forgot to change the unofficial internal names to the official names in the publication, the figure has been corrected.
Minor issues
All minor issues mentioned were changed according to reviewer’s suggestion, including the scaling of the axis.
Round 2
Reviewer 1 Report
Dear Authors,
Thank you for the revisions and replies to my comments. I am now happy with the manuscript.
Author Response
Dear Reviewer,
Thank you for reviewing our manuscript.
Reviewer 2 Report
I found very few mistakes that should be corrected:
line 339 - put an ending bracket after "farmers"
line 342 - remove bracket
line 361 - isotridecanol (with initial lower case; it is a rule for active substances used as PP - pesticides, wetting agents, etc)
line 415 - remove space between the number (0.01) and "%" (% is not a unit symbol; it is a fractional number with 100 in the denominator)
Author Response
Dear Reviewer,
thank you again for your commentss.
We changed the first letter of isotridecanol and removed the space between the numbers and the % symbol according to your suggestion.
The brackets in lines 139 to 142 have not been changed, but the entire sentence in the brackets has been moved to the main text. It is an explanation of how we determined the beginning of the weevil immigration. To avoid interrupting the flow of the text with this long explanation in the brackets, we have now decided to put it before this sentence.